# Estimations of Fracture Surface Area Using Tracer and Temperature Data in Geothermal Fields

**Anna Suzuki [1],\* , Fuad Ikhwanda [2], Aoi Yamaguchi [2] and Toshiyuki Hashida [2]**

[1]   Institute of Fluid Science, Tohoku University; 2-2-1 Katahira, Aoba-ku, Sendai, Miyagi 980-8577, Japan
[2]   Fracture and Reliability Research Institute, Graduate School of Engineering, Tohoku University;
      6-6-11-709 Aza-Aoba, Aoba-ku, Sendai, Miyagi 980-8579, Japan; fuad.ikhwanda@gmail.com (F.I.);
      aoi.yamaguchi@rift.mech.tohoku.ac.jp (A.Y.); hashida@rift.mech.tohoku.ac.jp (T.H.)
\*   Correspondence: Anna Suzuki: anna.suzuki@tohoku.ac.jp; Tel.: +81-22-217-5284

**Abstract:** Reinjection is crucial for sustainable geothermal developments. In order to predict thermal performances due to cold-water injection, a method was developed to estimate effective fracture surface areas (i.e., heat transfer areas). Tracer response curves at production wells are analyzed to determine flow rates and pore volumes, and the fracture surface areas are optimized by short-term thermal response curves. Because the method erases fracture apertures from the equation by combining mass and heat transfer equations, the fracture surfaces can be analyzed without assuming that the fracture shape is a parallel plate. The estimation method was applied to two geothermal field datasets: One involved an artificially created reservoir, and the other involved a naturally occurring reservoir. The estimated heat transfer areas are reasonable in the field geometries. Once the fracture surface area is estimated, the future temperature change and power generation can be predicted. This can provide a simple and quick method to design reinjection strategies.

**Keywords:** geothermal; injection; tracer; field data; flow paths; optimization

## 1. Introduction

Geothermal energy has been used to generate electricity and to provide hot water directly. Conventionally, hot water and steam in reservoirs, which originate from rain water, has been produced for geothermal utilization passively to avoid dry-out. Reinjection that returns water to a reservoir artificially is expected to maintain the pressure and the amount of water in reservoirs and to make the uses of the reservoirs sustainable. Because the injected water is usually cooler than the reservoir temperature, it is necessary to predict thermal performances due to cold-water injection.

Inter-well flow tests have been conducted to obtain important information of reservoir characteristics. These data are interpreted to build reservoir models. Most of commercial reservoir simulators are based on distributed parameter models (e.g., FE-FLOW [1] and TOUGH2 [2]). The distributed parameter models require flow and rock properties on each calculation cell in advance. Some of the models describe fractures implicitly by using porous blocks (e.g., MINC model [3]), while the others express fracture explicitly by using both fracture and porous blocks separately with fine grids [4–6]. In practical, the model consisting of fractures explicitly is expected to reproduce more realistic tracer response and thermal response curves. However, the approaches need huge computational cost to develop the reservoir models. There is also high uncertainty in determining the required parameters [7]. Because measurement data is insufficient at an early stage of development, a simplified approach can accelerate the development.

To simulate propagation of a cold front during reinjection, models consisting of a single fracture located in an infinite matrix has been used in the geothermal industry since the 1960s [8]. The simplified

approach is expected to have less uncertainty than the distributed parameter models. Robinson and Tester (1984) [9] and Robinson et al. (1988) [10] used analytical solutions for heat exchangers with one-dimensional flow in the "fracture" and no flow in the "matrix". They matched temperature histories of two EGS (Enhanced Geothermal System) projects by adjusting surface areas in a single fracture.

Pruess and Bodvarsson (1984) [11] have shown that the effective fracture aperture is the most influential parameter on the flow behaviors of cold front. Once the fracture aperture was determined, the fracture surface area for heat transfer was available in the referred model. Several researchers have estimated fracture apertures from laboratory and field tracer data with some success [12,13], but chemical tracers provide information of only water volume not interface surface area, which is not helpful to estimate temperature or flow-wetted surface area [12].

Simplified models assumed that the fracture aperture was perfectly uniform in the entire area. This assumption normally oversimplified complex reservoir structures. On the other hand, models accounting for several flow paths has been developed to interpret tracer recovery and been used successfully in various geothermal fields in Iceland and worldwide (i.e., [14,15]). The model can estimate properties of all flow paths, such as pore-space volume and dispersivity.

Based on the flow path model, Shook and Suzuki (2017) [16] proposed a method to determine flow properties for several flow paths uniquely, by using tracer and short-term thermal response data between a pair of an injection and a production wells. Their method removed the fracture aperture by combining mass and heat transfer equations and allows to estimate effective heat transfer areas for multi-channel flow. Their method is based on analytical solutions, which can be obtained by even spreadsheet prgrams (i.e., Excel). The method can analyze field data quickly and leads to reduce computational cost and to reduce uncertainty of optimizing large amount of input parameters in reservoir simulators.

In this paper, we summarized a series of estimation of heat transfer area and future prediction using tracer and short-term temperature decline curves based on the method from Shook and Suzuki (2017) [16]. Although the method was already proposed [16], they did not apply to real field data. In addition, in their model, flow rates and pore volumes in flow paths were estimated from tracer responses, and the surface area of the flow paths were estimated by fitting temperature curves. Ikhwanda et al. (2018) [17] modified the model to be able to cope with cases where the injected water is not fully recovered and used virtual number of flow paths to increase the degree of freedom. This study applied the model modified by Ikhwanda et al. (2018) [17] to two field data in order to investigate the applicability of the method. One data was obtained from an artificially created reservoir, and the other was obtained from a naturally occurring reservoir. The estimations were discussed by comparing with the field geometries.

## 2. Methods

Shook and Suzuki (2017) [16] developed a simple estimation method for fracture surface areas based on tracer response and short-term temperature decline curve. In their model, flow rates and pore volumes in flow paths were estimated from tracer responses, and the surface area of the flow paths were estimated by fitting temperature curves. Ikhwanda et al. (2018) [17] modified the model to be able to cope with cases where the injected water is not fully recovered. We used the model modified by Ikhwanda et al. (2018) [17]. A series of analysis procedures for estimating fracture surface area and designing injection rate are introduced below and shown in Figure 1.

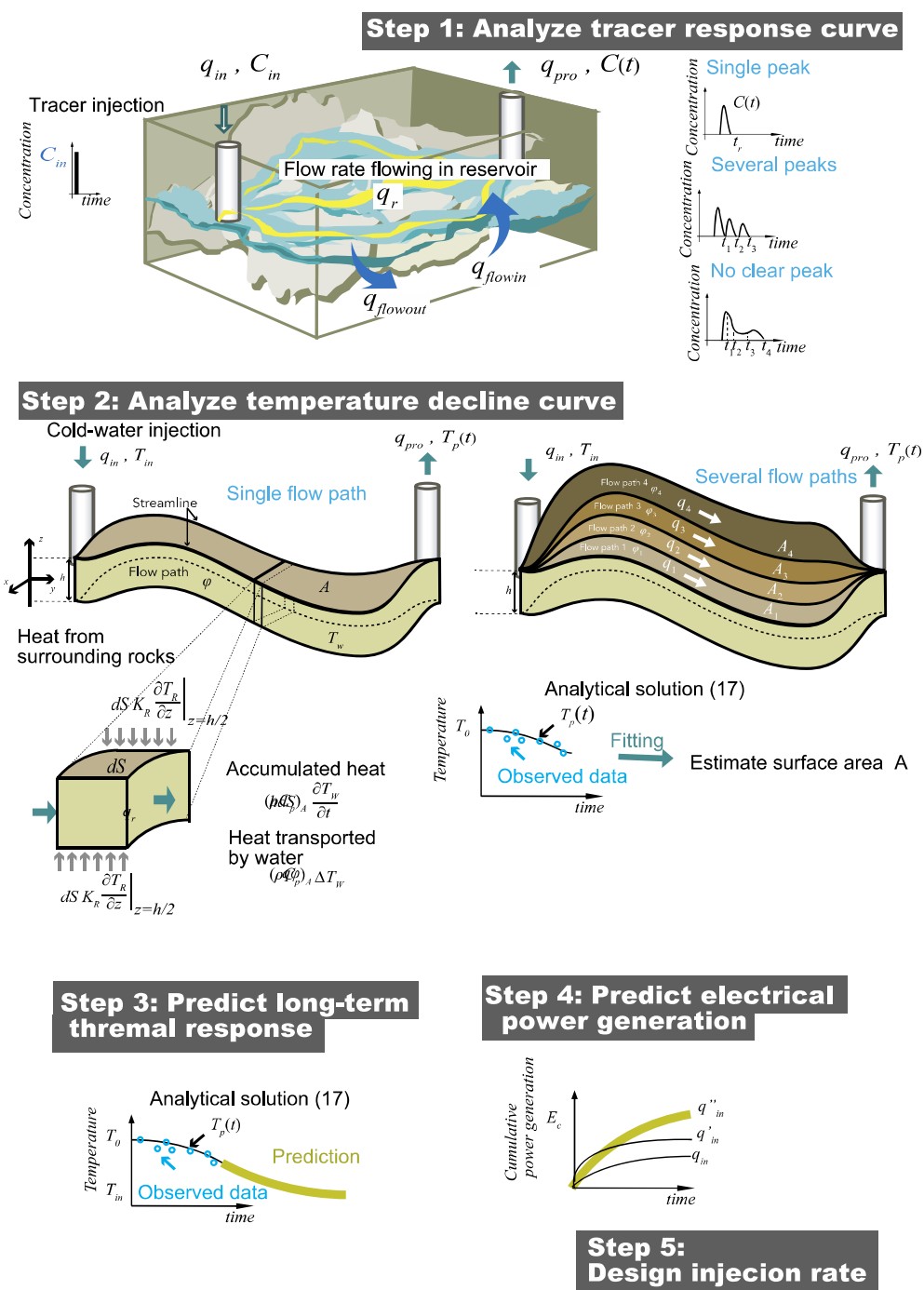

**Figure 1.** Workflow of procedure for estimating fracture surface area and predicting temperature decline and future power generation.

- Step 1

First, we assume that a reservoir consists with several flow paths of parallel plates sandwiched by low permeable rocks, and that the flow velocity within the reservoir is constant. Cold water is injected continuously. Pulse-like tracers are injected from an injection well with injected fluid, and a concentration distribution of the tracers are detected at an observation well. The temperature decline curve is observed at the production well when cold water is injected from the injection well.

The observed concentration of the tracers at a certain time is written in $c$ (t) [kg/s], and the recovery rate $f$ [-] of the tracers is:

$$f = \frac{1}{c_{in}} \int_0^{t_r} c(t)dt = \frac{c_r}{c_{in}}, \tag{1}$$

where $t$ [s] is time, $t_r$ [s] is the latest time at which the tracer has been detected, $c_{in}$ [kg/s] is the injection rate of the tracer, and $c_r$ [kg/s] is the cumulative concentration of the tracers. Considering the flow rate flowing in the reservoir $q_r$ [m$^3$/s], the injection flow rate $q_{in}$ [m$^3$/s], the production flow rate $q_{pro}$ [m$^3$/s], the flow rate flowing out from the reservoir to the outside $q_{flowout}$ [m$^3$/s], and the flow rate flowing into the reservoir from the outside $q_{flowin}$ [m$^3$/s], they are written in [8]:

$$q_r = fq_{in}, \tag{2}$$

$$q_{flowout} = (1-f)q_{in}, \tag{3}$$

$$q_{flowin} = q_{pro} - fq_{in}. \tag{4}$$

According to Equations (1) and (2), the flow rate flowing in the reservoir is multiplied by the cumulative concentration of the tracer over the injection concentration.

$$q_r = fq_{in} = \frac{q_{in}}{c_{in}}c_r. \tag{5}$$

If all of tracers migrate with the flow rate $q_r$ with the average travel time of $\bar{t}$, their product, $q_r\bar{t}$, can be considered as the pore volume $V_p$ [m$^3$] of the flow paths between the wells, and the following relationship is obtained according to Equation (5):

$$V_p = q_r\bar{t} = fq_{in}\bar{t} = \frac{q_{in}\bar{t}}{c_{in}}c_r. \tag{6}$$

Subsequently, tracers may pass through several different flow paths, and the several peaks in the tracer response may appear independently. Let us segment the tracer response with $n$ peaks into $n$ flow paths, respectively. If the peak is separately each other, each cumulative concentration can be written in

$$c_i = \int_{t_{i-1}}^{t_i} c(t)dt. \tag{7}$$

where the segmentation time is set to 0, $t_1, t_2, .., t_i, \ldots, t_n$. Given the recovery rate $f_i$ for the $i$-th flow path, the flow rates $q_i$ and void volumes $V_{pi}$ of the respective flow paths are as follows.

$$q_i = f_iq_{in} = \frac{q_{in}}{c_{in}} \int_{t_{i-1}}^{t_i} c(t)dt, \quad i = 1, 2, \ldots, n; \tag{8}$$

$$V_{pi} = q_i\bar{t}_i = f_iq_{in}\bar{t}_i = \frac{q_{in}\bar{t}_i}{c_{in}} \int_{t_{i-1}}^{t_i} c(t)dt, \quad i = 1, 2, \ldots, n. \tag{9}$$

Obvious sharp tracer peaks may appear discretely in some cases, but in many cases, response curves appear continuously. For convenience, even if there is no clear peak in a tracer response curve, we divide the continuous tracer response at certain time intervals (from $t_{i-1}$ to $t_i$) into $n$ flow paths. The number, $n$, is effective value to separate the tracer responses, which is not necessarily same as the actual number of flow paths in a reservoir. The flow rate of each flow path can be expressed using the

ratio of tracer concentration, and the void volume can be expressed using the ratio of the product of tracer concentration and observation time. From these assumptions, we can obtain the followings:

$$f = \sum_{i=1}^{n} f_i, \tag{10}$$

$$c_{total} = \sum_{i=1}^{n} \int_{t_{i-1}}^{t_i} c(t)dt = fc_{in}, \tag{11}$$

$$q_{total} = \sum_{i=1}^{n} q_i = q_r, \tag{12}$$

$$V_{ptotal} = \sum_{i=1}^{n} V_{pi}. \tag{13}$$

- Step 2

Optimal surface area of flow path $A$ is estimated by inverse analysis using a short-time temperature decline curve. Equation (14) shows a heat transfer equation of water flowing within a narrow fracture in a parallel plate [18].

$$\frac{h}{2}(\rho C_p)_A \frac{\partial T_W{}^{\psi}(S,t)}{\partial t} + \frac{q}{2}\rho_W C_{pW} \frac{\partial T_W{}^{\psi}(S,t)}{\partial S} = K_R \frac{\partial T_R{}^{\psi}(S,z,t)}{\partial z}\Big|_{z=h/2}. \tag{14}$$

where $T_w$ is the water temperature in a fracture, $T_R$ is the temperature in surrounding rocks, $S$ is the surface area, $t$ is time, and $z$ is the vertical coordinate axis to the fracture. The parameter $h$ is the fracture aperture, $q$ is flow rate in the fracture, $\rho$ is the density, and $C_p$ is the heat capacity. $(\rho C_p)_A = \varphi\rho_w C_{pw} + (1-\varphi)\rho_R C_{pR.}$, and $\varphi$ is a porosity. Note that the subscript $R$ means rock, subscript $w$ is water, and subscript $A$ is total of water and rock. The heat transfer of the surrounding rock in the vertical coordinate is given by the following equation.

$$\frac{\partial^2 T_R{}^{\psi}(S,z,t)}{\partial z^2} = \frac{\rho_R C_{pR}}{K_R} \frac{\partial T_R{}^{\psi}(S,t)}{\partial t} \qquad \left(z \geq \frac{h}{2}\right) \tag{15}$$

where $K_R$ is the thermal conductivity. From Equations (14) and (15) and the following boundary condition (16), an analytical solution (17) can be written in the following form.

$$T_W{}^{\psi}(0,t) = T_0 \quad t \leq 0, \tag{16a}$$

$$T_W{}^{\psi}(0,t) = T_t \quad t > 0, \tag{16b}$$

$$T_W{}^{\psi}(S,t) = T_W{}^{\psi}(S,h/2,t) \quad \forall t \text{ and } S, \tag{16c}$$

$$\lim_{z \to \infty} T_R{}^{\psi}(S,z,t) = T_0. \tag{16d}$$

$$T_P = T_0 - (T_0 - T_{in})\text{erfc}\left[\frac{A\sqrt{K_R(\rho C_p)_R}}{(\rho C_p)_w 2q\left(t - \frac{(\rho C_p)_A}{(\rho C_p)_w}\frac{V_p}{\varphi q}\right)^{\frac{1}{2}}}\right]. \tag{17}$$

where $T_0$ is the initial reservoir temperature, $T_P$ is production temperature, $T_{in}$ is the temperature of injected water. Since the heat transfer area of a flow path, $A$, is only unknown in the analysis solution (15), the value is estimated by fitting the analytical solution (17) to a thermal response obtained

from a field. We used the L-BFGS-B method from the python *scipy* package to minimize errors between the solution (17) and a field data.

　　Equation (14) assumes that there is a single flow path in a reservoir with a flow rate $q$. As same as Step 1, Let us consider $n$ flow paths in heat transfer equation. The temperature decline observed at a production well $T_p$ is considered to be the superposition of the thermal responses of each flow path. Denoting the temperature change of each flow path as $T_{p1}$, $T_{p2}$, ..., $T_{pn}$, the temperature change at the production well for multiple flow paths can be written in:

$$T_p(t) = \frac{1}{q_{pro}} \sum_{i=1}^{n} q_i T_{pi}(t).$$

(18)

$T_{pi}$ can be obtained by modifying Equation (17) as:

$$T_{Pi}(t) = T_0 - (T_0 - T_{in})\text{erfc}\left[\frac{A_i \sqrt{K_R(\rho C_p)_R}}{(\rho C_p)_w 2q_i\left(t - \frac{(\rho C_p)_T}{(\rho C_p)_w} \frac{V_{pi}}{\varphi_i q_i}\right)^{\frac{1}{2}}}\right].$$

(19)

where $T_{pi}$ is the temperature produced from *i*-th flow path, $A_i$ is the heat transfer area $V_{pi}$ is the pore volume, $\varphi_i$ is the porosity, and $q_i$ is the flow rate in the *i*-th flow path. The values of $V_{pi}$ and $q_i$ can be determined by Equations (12) and (13). By fitting thermal response curve with the solution (19), each surface area of each flow path is estimated, and the total surface area is calculated as the sum of each surface area.

$$A = \sum_{i=1}^{n} A_i.$$

(20)

　　Equation (14) includes porosity inside the fracture. Thus, the fracture part does not have to be a void space but a porous medium. In addition, if the fracture has rough surface, the model can simulate the heterogeneities by using the porosity. In that case, the surface area of the region where the injected water flows is estimated.

- Step 3

　　The estimated heat transfer areas for each flow path are substituted into analytical solutions (19) and (18) to predict long-term thermal responses with Equation (18).

- Step 4

　　The electrical power generation can be calculated by substituting the temperature decline curve from Step 3 into the following equations:

$$E(t) = q_{pro}C_{pw}(T_P(t) - T_{in}).$$

(21)

Using the power conversion efficiency $\eta = 6.9681 \ln(T_P) - 29.713$ [19], the cumulative power generation is given by:

$$E_C = \int E(T_P) \cdot \eta(T_P)\, dt.$$

(22)

- Step 5

　　To maximize the accumulated power production amount $E_c$ (22), the injection rate $q_{in}$ is determined. These Step 1 to Step 5 would help to design the optimal injection rate for sustainable development.

## 3. Validation

### 3.1. Simulation Setup

The estimation method was validated by simulating reservoir responses and calculating tracer and temperature histories using TOUGH2 (Pruess, 1991) [20]. A single well pair was set in a single vertical fracture set in non-fractured native rock and a single fracture surrounded by two damage zones, that is, three flow paths existed. We assumed the fracture permeability is a constant $1.0 \times 10^{-11}$ m$^2$ (~10 D). The half-length of the fracture was set to 99 m with a height of 75 m and an aperture of 0.02 m. The matrix width was 642.66 m to ensure semi-infinite media over the time scale of interest, with grid block sizes increasing by a factor of 2 away from the fracture to ensure numerical error was controlled. The wells are completed only in the fracture. Figure 2 shows a schematic of the model for TOUGH2. Thermal and physical properties of the rock were taken from the literature and are summarized in Table 1. The matrix is assumed to have no permeability.

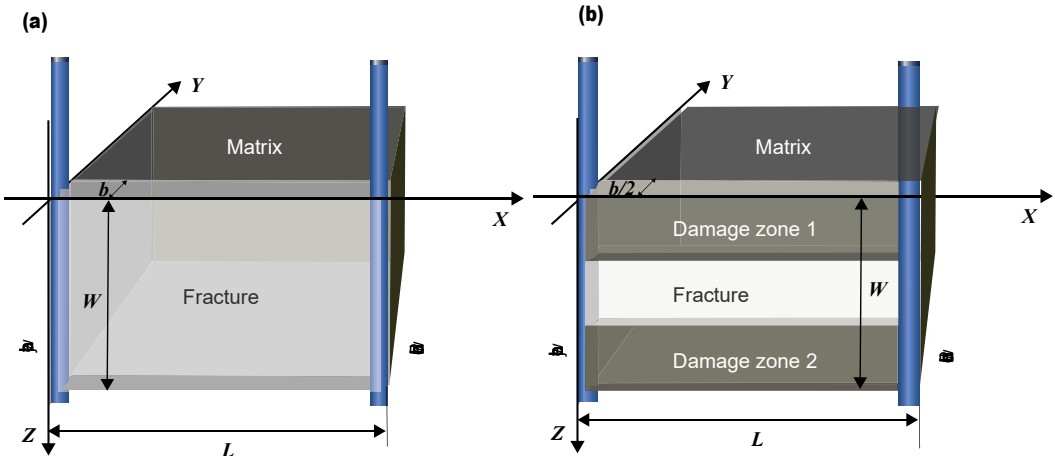

**Figure 2.** A schematic of the EGS reservoir: (**a**) uniform fracture and (**b**) three flow paths (a fracture surrounded by two damage zones).

**Table 1.** Simulation properties in TOUGH2.

| Parameter | Value | Parameter | Value | | |
|---|---|---|---|---|---|
| Rock Density (kg/m$^3$) | 2569 | | Single fracture | Three flow paths | |
| Rock heat capacity (J/kg °C) | 803 | | | DZ 1 | 0.75 |
| Rock thermal conductivity (W/m °C) | 2.569 | Porosity (-) | 0.9 | Fracture | 0.5 |
| Water density at 25 °C (kg/m$^3$) | 997.1 | | | DZ 2 | 0.95 |
| Water density at 200 °C (kg/m$^3$) | 864 | | | DZ 1 | 4 |
| Water heat capacity at 25 °C (J/kg K) | 4180 | Permeability ($10^{-12}$ m$^2$) | 10 | Fracture | 10 |
| Thermal conductivity at 200 °C (J/kg K) | 4510 | | | DZ 2 | 2 |
| Initial pressure (kPa) | 9800 | | | DZ 1 | 34.875 |
| Initial temperature (°C) | 200 | Pore volume (m$^3$) | 125.55 | Fracture | 23.25 |
| Flow rate (kg/s) | 2 | | | DZ 2 | 44.175 |
| Injection temperature (°C) | 25 | | | DZ 1 | 4650 |
| Model dimensions (m) | 99 × 75 × 643 | $A$ (m$^2$) | 13,950 | Fracture | 4650 |
| Grid dimensions | 33 × 15 × 3 | | | DZ 2 | 4650 |
| $\Delta X$ (m) | 3 | | | | |
| $\Delta Y$ (m) | 0.01, 0.02, 0.04, … | | | | |
| $\Delta Z$ (m) | 25 | | | | |

The initial temperature is 200 °C, and the initial pressure was set to 9800 kPa, about the hydrostatic pressure gradient assuming the fracture is at 1 km depth. At $t = 0$, water at 25 °C was injected for 1 hour, followed with 25 °C water with 10% tracer for one hour and then with 25 °C water without tracer for the balance of the simulation. All injection and production rates were 2 kg/s. For reasons of

symmetry we simulated a half-space solution, but the results showed below are for the full solution for a single well pair with surrounded rocks.

### 3.2. Numerical Results

The tracer responses and temperature histories for cases of (a) a uniform single fracture and (b) three flow paths were obtained from TOUGH2. Tracer responses are shown in Figure 3a,b. The tracer response for the case of the uniform fracture shows a bell-shaped distribution, while the curve for the three flow paths exhibits three peaks. We assumed that the total number of flow paths were unknown. Since the total number of flow paths was unknown, the number was considered a simulation parameter. As mentioned above, we divided the continuous tracer response at certain time intervals into the total number of flow paths. It does not matter what time the tracer response was segmented. In this study, the total flow rate was divided into the same amount of flow rates for each flow path so that each flow path had same flow rates. Once the flow rates and the segmentation times was determined for each flow path, each pore volume could be obtained from Equation (9) using each segmentation time.

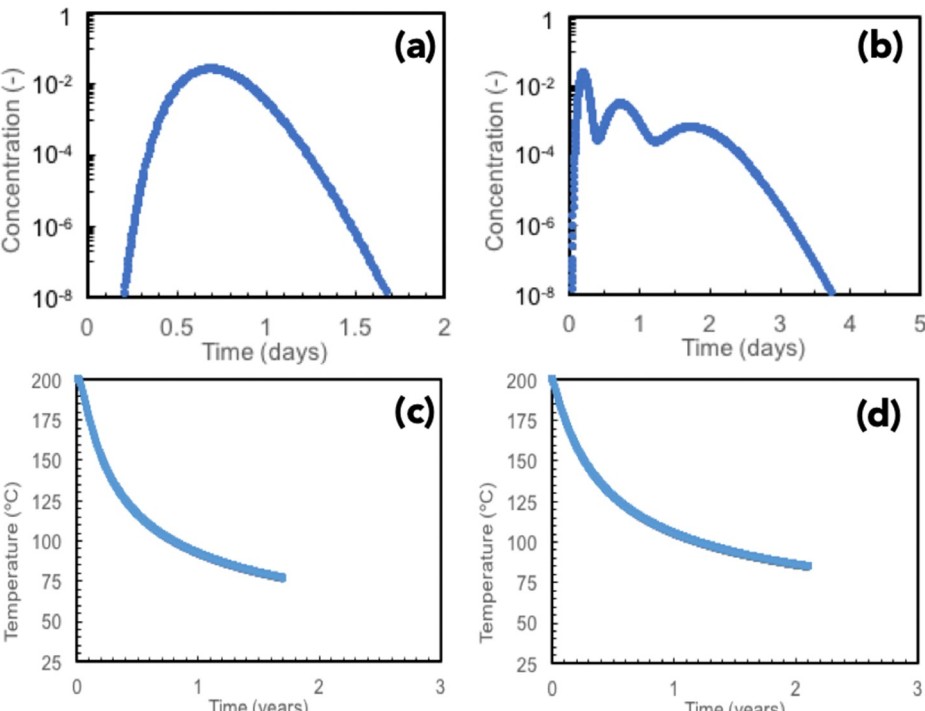

**Figure 3.** Tracer responses and temperature historires for (**a**) uniform fracture and (**b**) three flow paths, and temperature histories for (**c**) uniform fracture and (**d**) three flow paths.

Substituting the fractions of the flow rate $q_j$ and the fractions of the pore volume $V_{pj}$ into Equation (13), the solution of Equation (13) were compared with the temperature history simulated by TOUGH2 as shown in Figure 3c,d. We used the temperature decreasing by 70% of the difference between the initial temperature and the injection temperature. The fractions of the surface area $A_j$ were the optimized parameters. The objective function was the difference between the simulated and predicted values, and values for each fraction of surface area $A_j$ were optimized to minimize the objective function by the L-BFGS-B method in python function. For optimization, we set three different initial values of the fraction of surface area $A_j$ as 1000, 5000, and 10,000. Figure 4 shows the estimated results for fracture surface area. The true value of surface area was 13,950 m$^2$, plotted as "+". Each number of division used three initial values. The different colors describe the fractions of surface area. The estimated surface area is the total of the fractions.

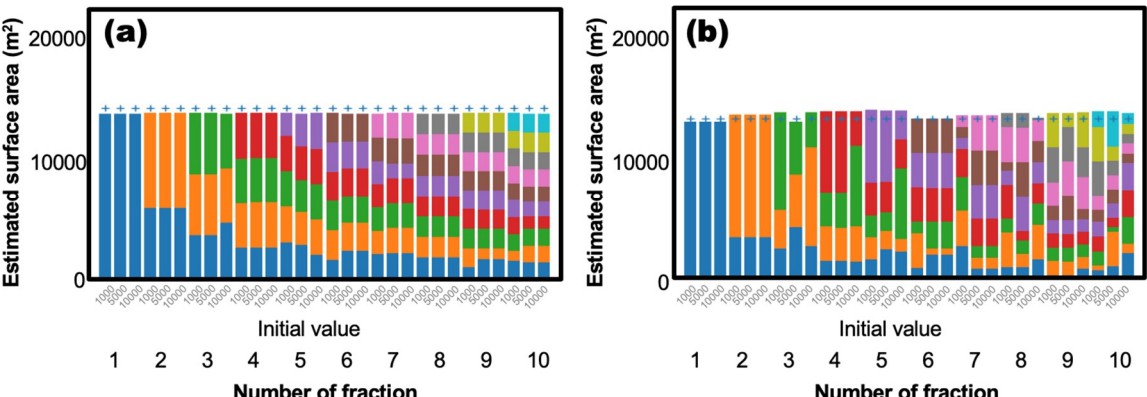

**Figure 4.** Estimated pore volumes for the case of (**a**) uniform fracture and (**b**) three flow paths. Exact answer is plotted as '+'.

When we changed the initial values for the fractions of surface area $A_j$, the fractions of surface area $A_j$ could vary as shown in Figure 4. However, the estimated total surface areas were almost same for the case of uniform fracture (Figure 4a) and gave the errors of around 2%. These results indicate that the optimization does not depend on the initial values for optimization and also the number of division of flow properties. In contrast, for the case of three flow paths, the estimated values differed for initial values for optimization and the number of division. Yet, the errors were in the range between 1% and 6%. This method led to the reasonable estimations of total surface area with premature temperature decline.

## 4. Field Data

This study used two field data and showed the applicability of the proposed method to estimate heat transfer area, as mentioned in Step 1 and Step 2 in Section 2, and then to predict future temperature change and power generation, as mentioned in Step 3-5.

### 4.1. Fenton Hill Phase I Reservoir

The Fenton Hill Phase I reservoir was the world's first man-made reservoir at Fenton Hill in the Jemez Mountains of northern New Mexico, which was designed to demonstrate the feasibility of creating and operating a prototype hot dry rock geothermal reservoir (EGS reservoir). In Phase I of the program conducted in the 1970s, a series of hydraulic fracturing and flow tests was conducted [21]. The conceptual model of the Phase I reservoir in Fenton Hill is shown in Figure 5 [22]. In Run Segment 2, a 75-day period of closed-loop operation was conducted. Water was pumped into the reservoir through well EE1 and recovered via a production well GT-2. The recovered hot water was cooled to 25 °C by the water-to-air heat exchanger and reinjected into well EE-1. a 200-ppm, 400-liter pulse of sodium-fluorescein dye was injected into well EE-1 wellhead, pumped down from well EE-1 and through the fractured region. The dye concentration in the produced fluid was monitored spectrophotometrically at the surface from the GT-2B wellbore. The tracer response was plotted in Figure 6a. The concentration was normalized by the sum of observed concentration (see [22]). The variation of temperature measured at a 2.6 km depth in GT-2B is depicted in Figure 6b. Several numerical models were developed to estimate the heat-transfer area of the reservoir. Tester and Albright (1979) [23] suggested that the heat transfer surface was about 8000 m$^2$.

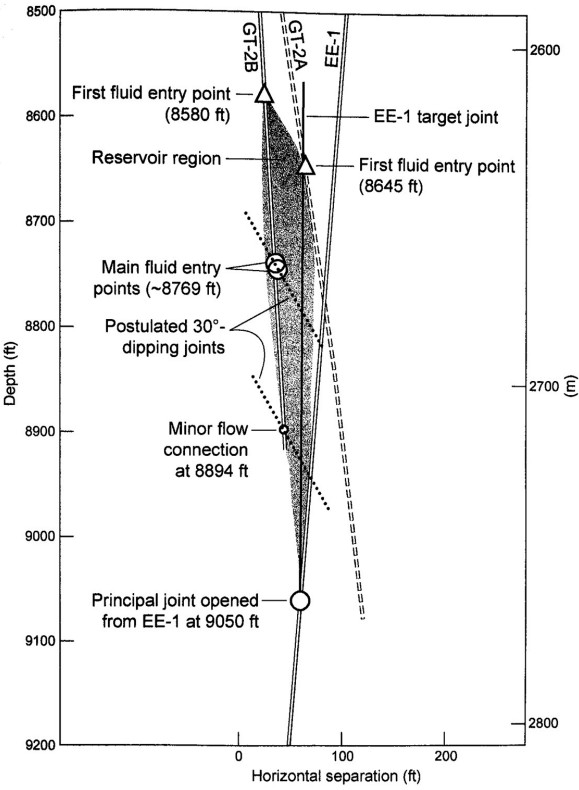

**Figure 5.** Conceptual model of the Phase I reservoir in Fenton Hill [22].

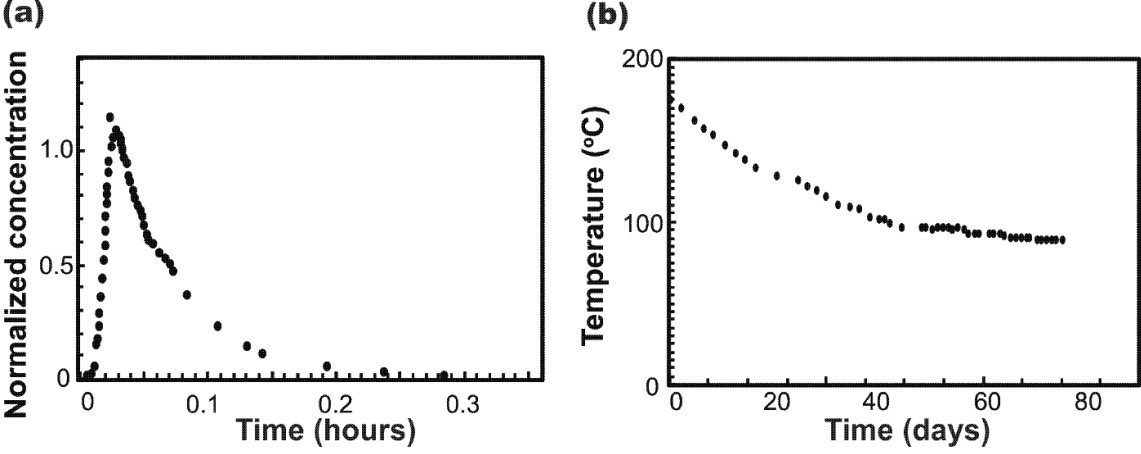

**Figure 6.** Field data from Fenton Hill HDR reservoir. (**a**) tracer response and (**b**) temperature decline curves [23].

## 4.2. Balcova Geothermal Field, Turkey

Balcova geothermal field is located on the shore of the Aegean Sea in Turkey and provides the largest geothermal district heating system in the country. The geothermal fluid is mainly used for public heating of municipal houses and greenhouses. The field view of Balcova field [24] is shown in Figure 7. A major 2-km long fracture, shown as the bold dashed line, in the reservoir is considered as the main region of fluid flow. Arkan et al. (2005) [25] provided estimated reservoir properties as listed in Table 2.

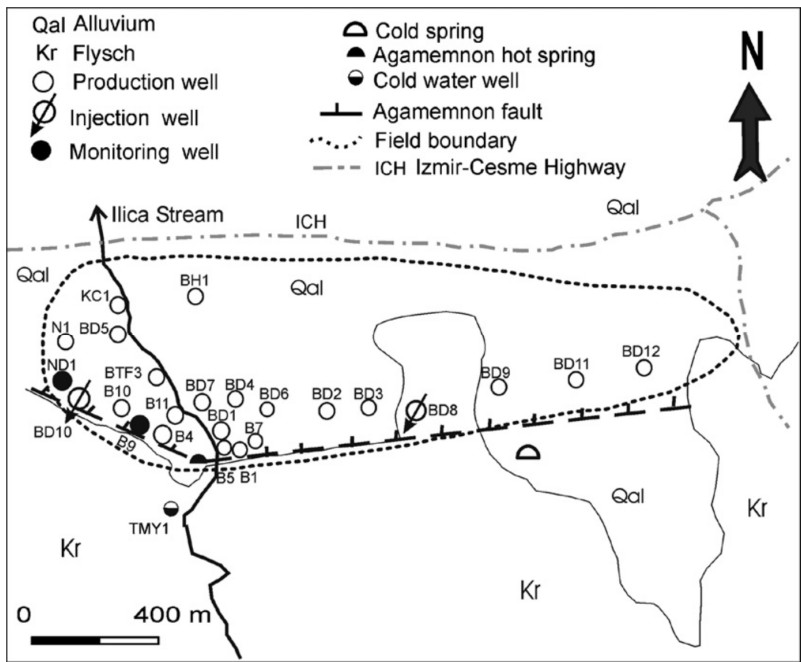

**Figure 7.** Field view of Balcova geothermal field [24].

**Table 2.** Properties of Balcova geothermal reservoir [25].

| Parameter | Most Likely | Min | Max |
|---|---|---|---|
| Porosity [-] | - | 0.002 | 0.070 |
| Rock specific heat [J/kg] | 0.92 | 0.80 | 1.08 |
| Rock density [kg/m$^3$] | 2750 | 2600 | 2850 |
| Rock temperature [°C] | 135 | 100 | 145 |
| Area [m$^2$] | $9.00 \times 10^5$ | $5.00 \times 10^5$ | $2.00 \times 10^5$ |
| Thickness [km] | 0.35 | 0.25 | 1.00 |
| Fluid density [kg/m$^3$] | 930.6 | 921.7 | 958.1 |
| Utilized temperature [°C] | 80 | - | - |
| Fluid specific heat [J/kg] | 4.18 | - | - |

In 2003, flow experiment was carried out in the Balcova geothermal field. Cold water at 60 °C was injected to well B9 and recovered at several production wells. Large portion of injected fluid was recovered in well B4. We used the data obtained at well B4 as the major connection to the injection well. Figure 8 plots the tracer response and the thermal response curve observed at well B4 in the Balcova geothermal field.

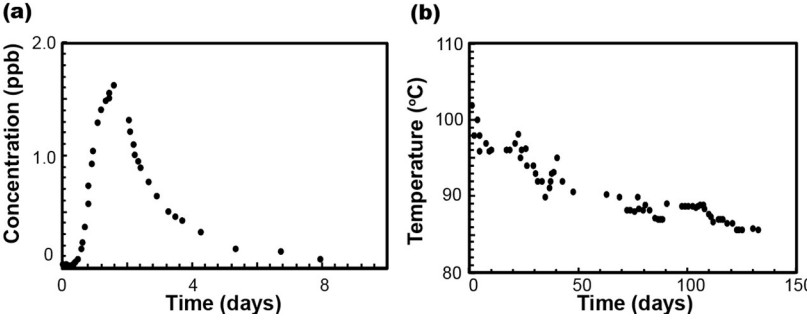

**Figure 8.** Field data from Balcova geothermal field. (**a**) tracer response and (**b**) temperature decline curves [24].

## 5. Results

### 5.1. Fenton Hill Geothermal Field, US

We analyzed the tracer response obtained from Fenton Hill Phase I reservoir (Figure 6a). The production rate $q_{pro}$ was 7.26 L/s, the recovery rate $f$ was 69%, the injection time was 38 s, respectively (modified from [23]). The tracer response had a sharp peak and a long tailing as shown in Figure 6a, which was difficult to detect the number of flow paths existing in the reservoir. Since the total number of flow paths was unknown, the number was considered a simulation parameter. The total number of flow paths was varied from one to nine and compared their reproductivity of the thermal response. As mentioned above, we divided the continuous tracer response at certain time intervals into the total number of flow paths. It does not matter what time the tracer response was segmented. In this study, the total flow rate was divided into the same amount of flow rates for each flow path so that each flow path had same flow rates. Once the flow rates and the segmentation times was determined for each flow path, each pore volume could be obtained from Equation (9) using each segmentation time. Figure 9 shows the pore volume for each flow path with different numbers of flow paths. The different colors in the column depicts the fractions of the pore volume for each flow path. If the flow media is homogeneous, the ratio of fractions of flow rate and pore volume should be same for each number of flow path. On the other hand, if the flow media is inhomogeneous, such as fracture-matrix systems, large portion of water flows quickly in preferential flow path (i.e., fractures) and small portion of water flows slowly in matrix. For the case where the total number of flow paths was two in Figure 9, the ratio of pore volume had smaller portion of the 1st path (blue) and larger portion of the 2nd path (yellow brown). This suggests that fluid in the 1st path flows faster than fluid in the second path. The 1st flow path represents a preferential flow path.

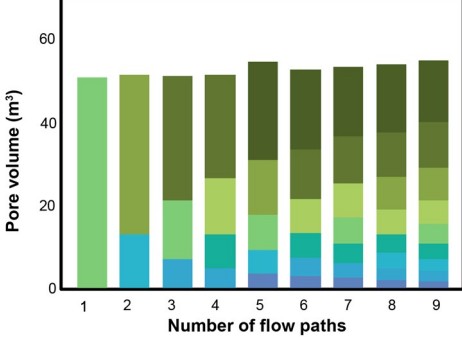

**Figure 9.** Pore volume calculated from tracer response curve from Fenton Hill HDR reservoir.

Preferential flow paths should be permeable and have higher porosity. Flow paths formed by a fracture only may be considered to have a porosity of 1.0, while the fracture porosity may be better to think to be smaller than 1.0 due to the roughness of fracture surfaces. In this way, the determination of the porosity remains uncertain. Thus, we changed the porosity for each flow path gradually and set the values in Equation (19). The porosity for *i*-th flow path was given using the following equation systematically:

$$\phi_{frac-i} = \phi_{frac-max} - \frac{i}{1+N}\left(\phi_{frac-max} - \phi_{frac-min}\right), \tag{23}$$

where $\varphi_{frac}$ is the fracture porosity, $N$ is the total number of flow paths, and $i$ is the index of the flow paths. The possible range of the porosity was from 0 to 1.0. The intermediate values in a possible range for each flow path was given by using Equation (23).

Temperature profiles of Fenton Hill Phase I reservoir (Figure 6b) was used to estimate surface area. Figure 10a depicts the residual errors for each optimization with different initial values and different

number of flow paths. The residual errors were the mean absolute errors between the analytical solution and field data. We varied the number of flow paths and the initial values and optimized to estimate the surface area by minimizing the residual errors between the observation data and the calculated curve from Equations (18) and (19). The smallest residual was 56.38 °C when the number of flow paths was nine and the initial value was 100 m². In this case, the estimated surface area was 6013 m². As shown in Figure 10, the residual errors were almost same except the cases of four flow paths with the initial value of 100 m². This result was regarded as an outlier of optimization. The average of heat transfer area calculated without the outlier was 5485 m². The errors of estimated value for each optimization from the average 5485 m² was in the range between −8.6% and 18.2%. The results suggests that the optimization does not be affected by the initial values and the number of flow paths. The value 5485 m² was smaller than the estimation of 8000 m² from Tester and Albright (1979) [23]. Their model considered the fracture was void space, that is the porosity was 1.0. When we used higher porosity, the estimation became larger and close to the estimation of 8000 m². From this point, we need to be careful to determine the porosity. We will discuss the sensitivity of porosity not only for estimation of surface area but also for future prediction of production in next study.

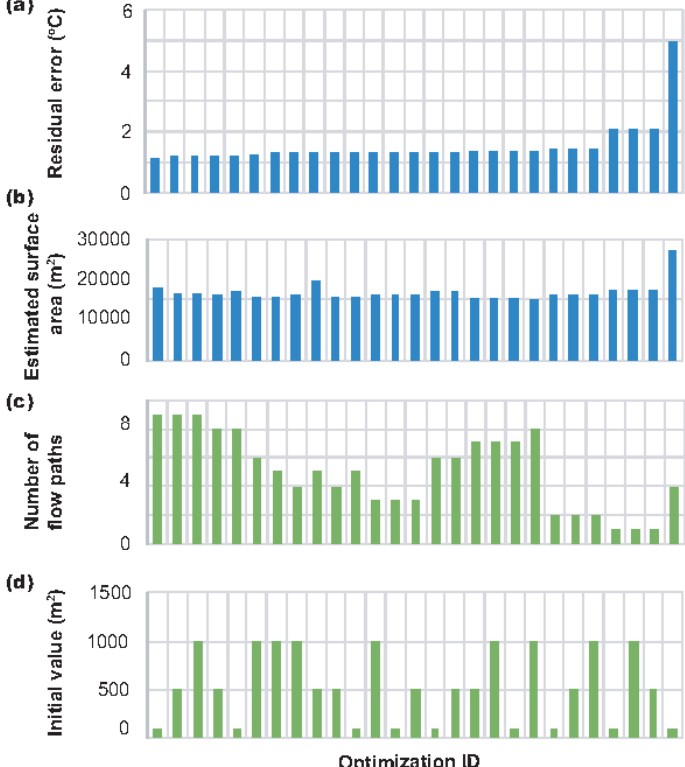

**Figure 10.** Optimization of surface area between well EE-1 and GT-2B in Fenton Hill HDR reservoir. (**a**) residual errors between field data and calculated curve, (**b**) estimated heat transfer area, (**c**) number of flow paths, and (**d**) initial values for optimization. The optimization ID was arranged in order from the smallest residual errors.

The smallest residual error was provided when the number of flow paths was nine and the initial value was 100 m². The fitting result was plotted in Figure 11. The calculated curve has a good agreement with the Fenton Hill Phase I reservoir. If one assumes that a single flowing joint makes an almost direct connection between well EE-1 and well GT-2B and that the heat transfer area is a roughly rectangular joint or ellipse joint with an inlet at 2758 m (9050 ft) in EE-1 and outlet at 2673 m (8769 ft) in GT-2B, the height difference is 86 m (281 ft) (see Figure 5). The estimated heat transfer areas of a rectangular and an ellipse for nine flow paths are described in Figure 12b,c. The color describes the portion of each flow path from 1st at porosity of 1.0 (blue) to 9th at porosity of 0.002 (dark brown).

The estimated length of one side of rectangular and ellipse were 70.7 m and 90.1 m, respectively. Figure 12b,c suggests that the estimated sizes of heat transfer area are reasonable. As mentioned above, the total number of flow path is a simulation parameter, which does not describe the actual flow path. Let us reduce the number of flow paths. The estimated heat transfer areas of a rectangular and an ellipse for a single flow path are described in Figure 12d,e. The one sides of rectangular and ellipse were 68.7 m and 87.5 m, respectively. There are less differences from the estimated sizes of nine flow paths (Figure 12b,c). Therefore, the thermal response in Fenton Hill Phase I reservoir can be reproduced with an assuming a single flow path, which suggests that a simple single fracture was created by artificial water stimulation in Phase I reservoir in Fenton Hill field and that the flow behavior in the reservoir can be considered homogeneous.

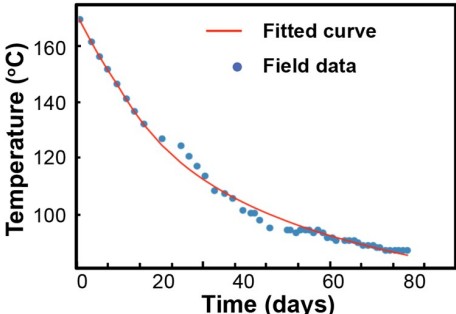

**Figure 11.** Curve fitting to temperature decline curve from Fenton Hill Phase I reservoir.

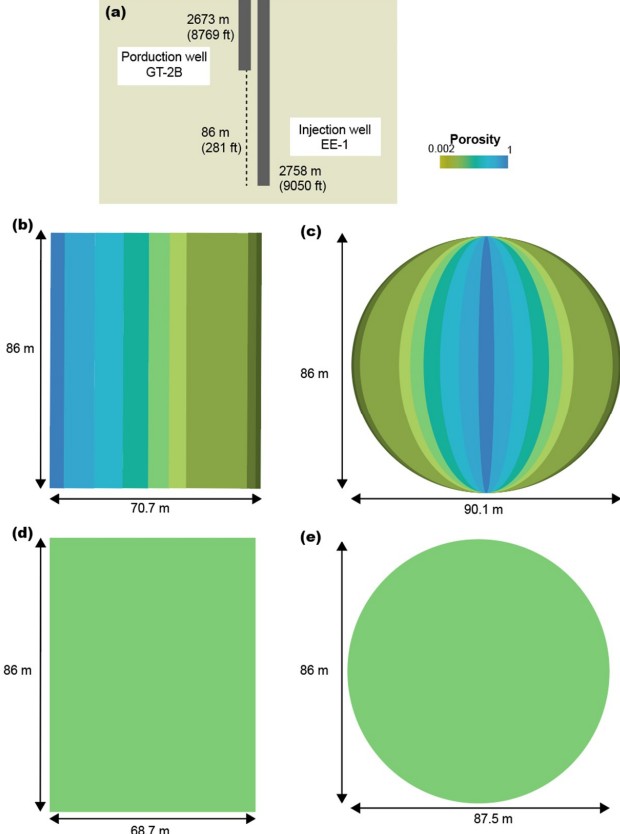

**Figure 12.** Estimated results of heat transfer area between well EE-1 and well GT-2B in the Fenton Hill Phase I reservoir. (**a**) Schematic of well position, (**b**,**c**) estimated surface areas with nine flow paths and (**d**,**e**) estimated surface area with two flow paths, assuming the area is (**b**,**d**) rectangular or (**c**,**e**) circle, respectively.

*5.2. Balcova Geothermal Field, Turkey*

　　We also analyzed the tracer response of Balcova field data [24] as shown in Figure 8a. As well as analyzing the data from Fenton Hill, the total number of flow paths was unknown. We varied the number of flow paths from one to nine. Based on the field observation [25], the reservoir porosity was estimated from 0.002 to 0.07, which mainly represented matrix porosity. The possible maximum porosity was set to 1.0, the minimum value was determined from the field observation (i.e., 0.002).

　　Figure 13 shows the pore volume for each flow path with different numbers of flow paths. The different colors in the column depicts the fractions of the pore volume for each flow path. Temperature profiles of Balcova field [24] (Figure 8b) was used to estimate the heat transfer area. We varied the number of flow paths and the initial values and conducted the optimization by minimizing the residual errors between the observation data and the calculated curve from Equations (18) and (19). Figure 14a depicts the residual errors for each optimization varying the initial values and the number of flow paths. The smallest residual error was 57.09 °C when the number of flow paths was nine and the initial value was 100 m$^2$. In this case, the estimated heat transfer area was 13,475 m$^2$. As shown in Figure 14, it can be seen that the larger the number of flow paths, the smaller the residual errors. This trend seems to be stronger than the result of Fenton Hill (see Figure 10).

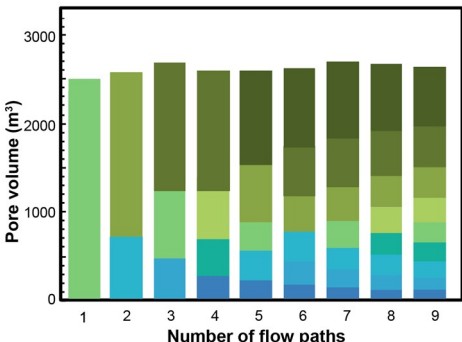

**Figure 13.** Pore volume calculated from tracer response curve from Balcova geothermal field.

　　The fitting result when the number of flow paths was nine and the initial value was 100 m$^2$ was plotted in Figure 15. The calculated curve has a good agreement with the Balcova field data. In Balcova field, the distance between well B4 and B9 was 114.3 m. Well B4 has a depth of 125 m, while well B9 has a depth of 48 m. The linear distance between two well was 137.8 m. Let us assume the heat transfer area is simply a rectangular or an ellipse. The estimated heat transfer area with nine flow paths are described in Figure 16b,c. The color describes the portion of each flow paths from 1st at porosity of 1.0 (blue) to 9th at porosity of 0.002 (dark brown). The lengths of one side of rectangular and ellipse were 97.8 m and 124.5 respectively. Figure 16b,c suggests that the estimated sizes of heat transfer area are reasonable.

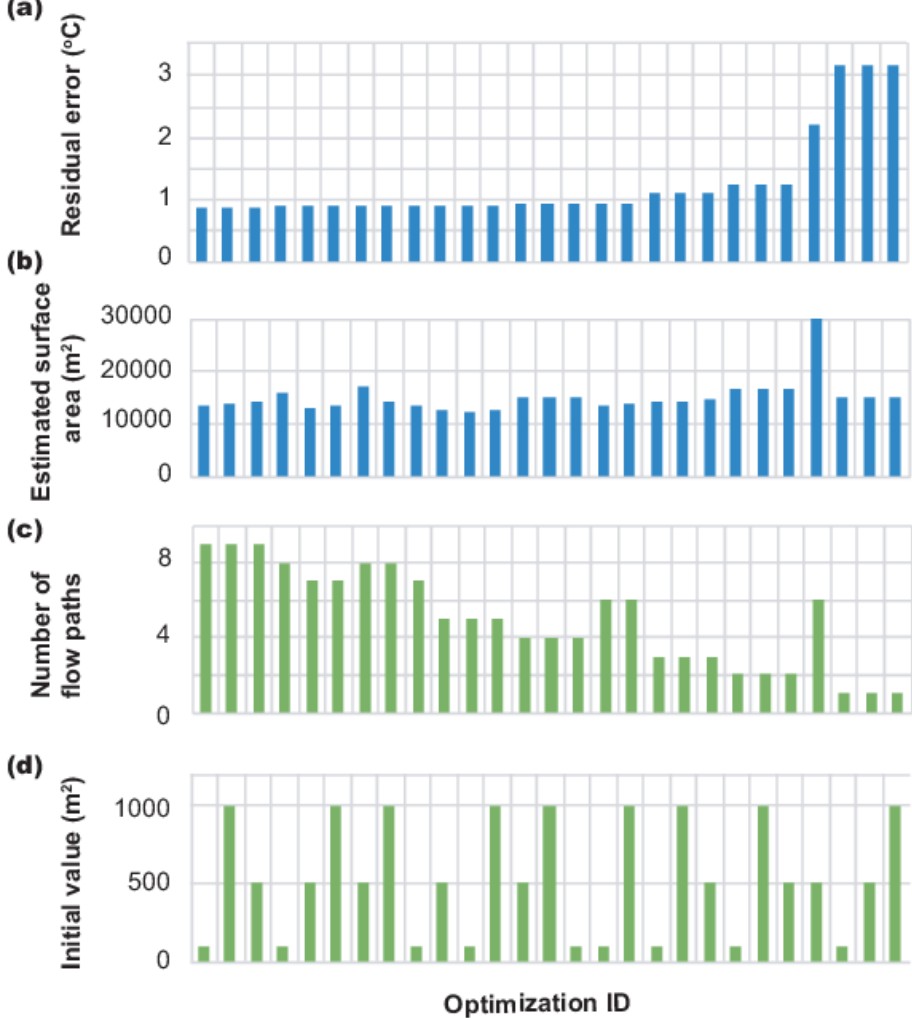

**Figure 14.** Optimization of surface area between well B4 and B9 in Balcova geothermal field. (**a**) residual errors between field data and calculated curve, (**b**) estimated surface area, (**c**) number of flow paths, and (**d**) initial value for optimization. The optimization ID was arranged in order from the smallest residual errors.

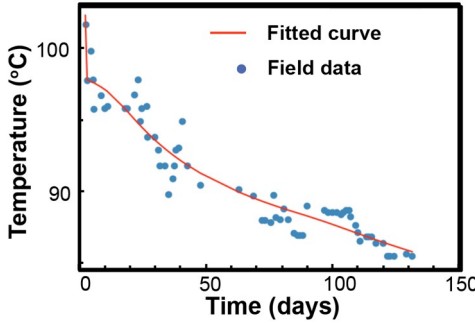

**Figure 15.** Estimated results of heat transfer area in the Balcova reservoir.

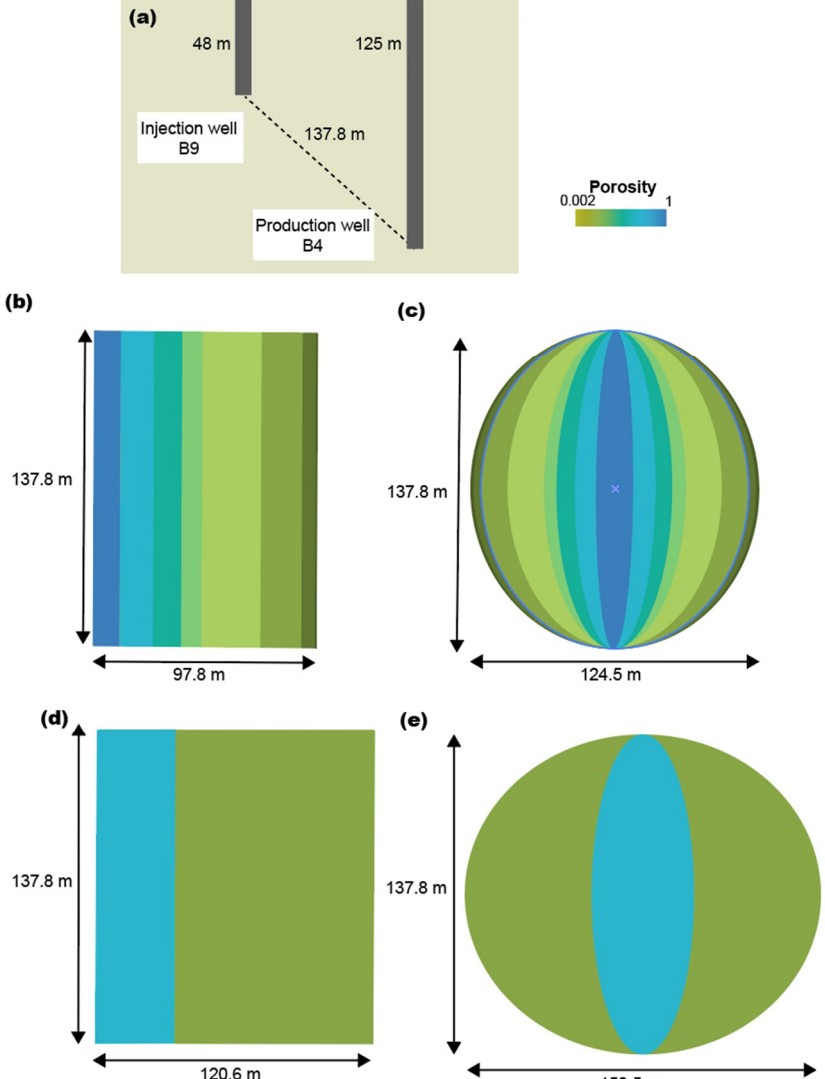

**Figure 16.** Estimated results of heat transfer area between well B4 and B9 in the Balcova geothermal field. (**a**) Schematic of well position, (**b,c**) estimated surface areas with nine flow paths, and (**d,e**) estimated surface area with two flow paths, assuming the area is (**b,d**) rectangular or (**c,e**) circle, respectively.

As shown in Figure 14, the optimization with a flow path generated very large residual errors. The case of six flow paths with the initial value of 500 m² could not provide adequate optimized value. Except these cases, two flow paths were the smallest number of flow paths. The estimated heat transfer area with two flow paths are described in Figure 16d,e. The rectangular and ellipse with two flow paths were larger than the results with nine flow paths. In addition, the portion of estimated surface area for 1st flow path (light blue) and for 2nd flow path (yellow brown) are different. This indicates that the thermal response cannot be described by considering two flow paths and that Balcova geothermal field consists of multiple flow paths with different flow patterns. Balcova geothermal field is a naturally occurring geothermal field. The fluid flow is controlled by natural rocks and faults. It is considered that more complicated fracture networks are formed in Balcova geothermal field than the artificial fracture in Fenton Hill Phase I reservoir.



### *5.3. Future Prediction*

In previous Sections 5.1 and 5.2, we estimated the fracture surface area from two field data. Once the fracture surface area is estimated, the subsequent long-term temperature and power production can be easily predicted. Here, an example of future prediction is shown by using Fenton Hill data.

Figure 17 shows the future prediction based on Fenton Hill data. The temperature change was predicted using the estimated fracture surface area, which is plotted as the blue dot in Figure 17a. In the field experiment, water was injected at 25 ° C with flow rate of 10.51 kg/s. The fitted curve and the long-term prediction of temperature change is shown in actual line in Figure 17a. We assumed that the injection flow rate was changed to 5 kg/s and 20 kg/s 74 days after the last measured field data. The temperature response for 5 kg/s and 20 kg/s are shown in dot line and dashed line, respectively. It can be seen that the larger the injection flow rate, the faster the temperature drop, while the smaller the injection flow rate, the milder the change. Figure 17b shows the cumulative power production calculated using Equations (21) and (22). Note that when the calculation formula of power conversion efficiency became a negative value, the value was calculated as 0. The recovery rate was fixed, and the production rate was changed according to the injection rate. From Equation (21), the larger the production flow, the greater the power for power production. On the other hand, the power production in Equation (21) is also a function of temperature, and if the temperature decreases, the power production also decreases. As shown in Figure 17a, When the injection rate was 20 kg/s, the temperature drops immediately, and no more power can be produced. Comparison between the cases for injection rate of 10.51 kg/s and of 5 kg /s revealed that the case for 10.51 kg/s increased power production at the beginning, but the case for 5kg/s can produce more electricity in the long run.

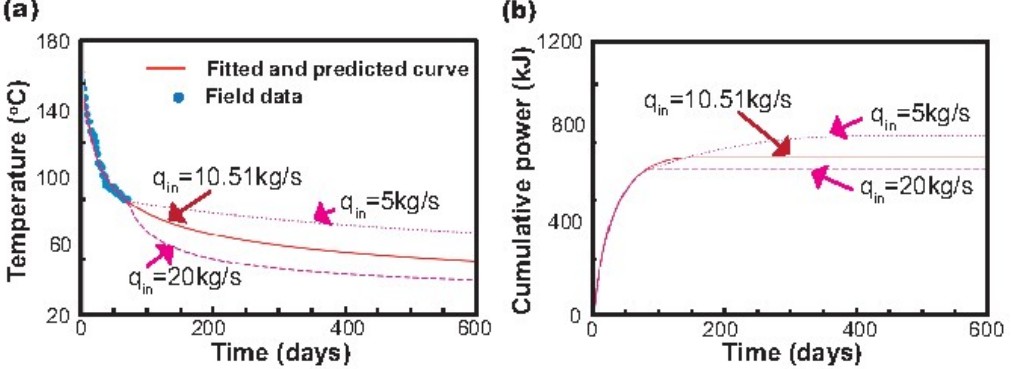

**Figure 17.** Future prediction by using Fenton Hill data. (**a**) Temperature declines for different injection rates and (**b**) cumulative power for different injection rates.

In the Fenton Hill experiment, cold-water injection was carried out until the temperature dropped considerably. If actual fields conduct in the same way, it would take a lot of time to recover the temperature because the thermal recovery is due to heat conduction, and it would have a large influence on field management. In the actual field, it is desirable to apply the method we proposed at a slightly earlier stage to control the injection rate and to make a sustainable design. Our method can be helpful to make better strategies at early stage for sustainable development.

### 6. Discussion and Conclusions

We used the number of flow paths as a fitting parameter in this method. Even if the structure is complex, increasing the number of flow paths as a fitting parameter increases the degree of freedom of optimization, so that it is possible to find an optimum value that will fit complex results well. The flow paths used in this model represent not actual flow paths but virtual ones. Actually, the number of flow paths is not necessary to be characterized. The number as the input parameter expresses how many patterns of flow exist from the injection well to the production well. The flow patterns that can

be recognized by tracer responses is limited. These limited parameters make the optimization easy. The analysis can be done quickly. Technically, spreadsheet software (i.e., Excel) can be used for this analysis. Thus, field operators do not prepare complicated numerical simulators and can easily try this method.

Conventional simplified models assumed that a fracture is rectangular [11,12] and estimated the fracture aperture for future prediction based on tracer response [26–28]. However, fractures are not always rectangular shapes, and only tracer responses cannot answer the relationship between the fracture aperture and the surface area. Their assumption includes discrepancy with actual structures. In this study, we erased the fracture aperture from the equation and avoid assuming the fracture is rectangular. This is a new approach to predict future production based on tracer and thermal responses.

One of weak points of this method is to require measurement of changes in tracer and thermal response. Because the cooling of reservoir may cause a decrease in steam production, field operators want to avoid as much as possible [29]. In addition, it takes time to wait until the thermal response changes. Kocabas and Horne (1990) [30] proposed thermal injection backflow tests to estimate the heat transport parameters using a well. This is one of method, which we do not have to wait until we obtain the temperature changes at a production well. However, the area between the wells is not always known from the method, and the results could be ambiguous.

Our method only estimates fracture surface area between wells, which cannot tell spatial distributions of fractures or temperature distributions. Thermally degrading chemical tracers has a potential to characterize temperature distributions between wells [31–33]. Time-lapse electrical resistivity tomography (ERT) offers the possibility of imaging noninvasively subsurface transport [34,35]. These technologies are not established yet, but when combined with these technologies, more information could be obtained from the tracer response. Our simplified method to analyze tracer responses can be helpful to characterize the reservoir structures.

**Author Contributions:** Conceptualization, A.S.; methodology, A.S.; software, A.S. and F.I.; validation, A.S. and F.I.; formal analysis, F.I. and A.Y.; data curation, F.I. and A.Y.; writing—original draft preparation, A.S. and F.I.; writing—review and editing, T.H.; visualization, A.S. and F.I.; supervision, T.H.; project administration, T.H.; funding acquisition, A.S. and T.H.

**Funding:** This work was supported by the Japan Society for the Promotion of Science under Grant-in-Aid for Young Scientists (A) (JP17H04976) and under Grant-in-Aid for Challenging Research (Exploratory) (JP17K19084), whose supports are gratefully acknowledged. Part of the work was carried out under the Collaborative Research Project of the Institute of Fluid Science, Tohoku University.

**Acknowledgments:** The authors thank Mike Shook for his helpful discussions concerning the estimation method. We also thank Niyazi Aksoy and Jefferson W. Tester for providing the field data used in Figures 6 and 8.

**Conflicts of Interest:** The authors declare no conflict of interest. The funders had no role in the design of the study; in the collection, analyses, or interpretation of data; in the writing of the manuscript, or in the decision to publish the results.

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
