# Peer review of "Estimations of Fracture Surface Area Using Tracer and Temperature Data in Geothermal Fields"

_geosciences, doi:10.3390/geosciences9100425_

Round 1

Reviewer 1 Report

The research is highly interesting and well written, and therefore accepted for publication.

Author Response

We thank you for carefully reading our manuscript, and for your useful comments. In response another reviewer’s comments, we had changed the title, which intended to convey the specific contents and revised the manuscript. We believe our manuscript satisfies your comments and hereby resubmit for your further consideration.

Reviewer 2 Report

Major comments:
Title: it is not clear if it it for any type of geothermal fields or only fractured rock
Abstract: The same ambiguity as in the title. Please make it more specific

Introduction: The literature review should be extended. Some of the statement requires references (e.g. Page 1 line 30-31)

Method: A figure presenting the method is required. It is also not clear which part of the method is new
Method: A verification study utilising a simple synthetic model would help readers to better understand the method. Please consider adding a verification section.

Results: before introducing the field model results, studying simple synthetic simple models (with, for example, parallel multiple-fracture) and varying different parameters (e.g. fracture spacing, fracture aperture, ..) should be added.

Results: Line 202-204. The authors proposed a model for fractured systems. I wonder how the method can be now applied for a reservoir without fractures (homogeneous media).

Results: The issues related to fracture roughness must be elaborated.
Results: Adding a comparison between this method and other analytical methods will be valuable

Results: I strongly recommend the authors to include the prediction part of the research (as mentioned in the conclusion) also in this manuscript.

Minor comments:
Figure captions are very short and not informative.
Residual error - how is it calculated?
Figure 2: How concentration is normalised?

Author Response

We thank you for carefully reading our manuscript, and for your useful comments. In response your comments, we had changed the title, which intended to convey the specific contents. We believe our manuscript satisfies your comments and hereby resubmit for your further consideration. The detailed response to you is as follows.

1.

Title: it is not clear if it it for any type of geothermal fields or only fractured rock

Abstract: The same ambiguity as in the title. Please make it more specific

As you pointed out, the title and abstract were revised to make it more specific.

Introduction: The literature review should be extended. Some of the statement requires references (e.g. Page 1 line 30-31)

Extended the literature review in Introduction.

3.

Method: A figure presenting the method is required. It is also not clear which part of the method is new.

Added the figure in Figure 1. The new point from previous studies were mention around L78.

4.

Method: A verification study utilising a simple synthetic model would help readers to better understand the method. Please consider adding a verification section.

Results: before introducing the field model results, studying simple synthetic simple models (with, for example, parallel multiple-fracture) and varying different parameters (e.g. fracture spacing, fracture aperture, ..) should be added.

Added the verification section in Section 3. The verification was conducted with a simple fracture and three flow paths with different permeability.

5.

Results: Line 202-204. The authors proposed a model for fractured systems. I wonder how the method can be now applied for a reservoir without fractures (homogeneous media).

6.

Results: The issues related to fracture roughness must be elaborated.

Commented in L264. Eq. (14) includes porosity inside the fracture. Thus, the fracture part does not have to be a void space but a porous medium. In addition, if the fracture has rough surface, the model can simulate the heterogeneities by using the porosity. In that case, the surface area of the region where the injected water flows is estimated.

7.

Results: Adding a comparison between this method and other analytical methods will be valuable.

We compared with other analyses and discussed the advantage and disadvantage of our method from L702.  Using measurement data could be valuable, which was mentioned from L716.

8.

Results: I strongly recommend the authors to include the prediction part of the research (as mentioned in the conclusion) also in this manuscript.

We provided the prediction results in Sceion 5.3.

9.

Minor comments:

Figure captions are very short and not informative.

Added more information.

10.

Residual error - how is it calculated?

In the previous version, we used absolute errors. Instead, mean absolute error was used, which was mentioned in P12.

11.

Figure 2: How concentration is normalised?

Added the explanation in P10.

Reviewer 3 Report

Review of the manuscript geosciences-533289 “Estimations of Heat Transfer Area Using Tracer and Temperature Data in Geothermal Fields” by Anna Suzuki, Fuad Ikhwanda, Aoi Yamaguchi and Toshiyuki Hashida.

This is an interesting paper that deals with the use of a method developed to estimate effective heat transfer areas. The authors have a good dataset from two different geothermal fields. By using tracer and temperature data, they obtain estimations about the heat transfer area. I consider that the results are adequately supported by the data. In my opinion, the paper results of interest for potential readers and consequently it could be accepted after a moderate revision is performed by the authors. More details are given below:

- The major issue of the manuscript is the lack of a Discussion section that must be compulsory in any scientific paper. Although the methodology exposed and the results described seem to be correct, it is important to discuss the results. This section must include a description of the novelty of the method in relation to previous works, as well as to describe the weak and strong points of the proposed method. What are the benefits of this method in relation to previous ones? What are the limitations encountered? How different are the results in the two geothermal areas compared with previous works? By answering these questions, the scientific significance of the manuscript will be increased.

- Apart from this, I have some comments about the results. It is surprising that, using data form two very different geothermal systems, one artificial and another one natural, the method provides exactly the same best fit for both of them. In both cases, the best fit is obtained when the number of flow paths is nine and the initial value was 100 m2, and both of them provides the same smallest residual error of ~50º-60º C temperature. It looks like the method has a certain tendency to obtain the same fit independently of the initial model. This is something that should be commented in the Discussion section proposed.

These observation results even more surprising when the authors state that, despite the fact that the best fit obtained for Fenton Hill field corresponds to a number of flow paths of nine, it is finally estimated that the best solution for this geothermal field is a single flow path, although this model has considerably greater (100º C, about twice greater) residual errors by looking at the graphs of figure 6. This is interesting because one can have serious doubts about the results of a method that assigns the lower errors to models that are not the best fit to the geothermal field characteristics, making difficult to decide how to select the best solution. This is controversial and should be clearly justified and exposed in the Discussion.

Last, just some minor corrections to be made:

 -          At Table 1, the thickness value (6th row) ranges from 0.35 to 1.00 mm. Such an extremely small value is impossible, and it must be an error that has to be fixed.

-          At figure 3, the labels are too small and very difficult to read. Please enlarge the figure.

Author Response

We thank you for carefully reading our manuscript, and for your useful comments. In response to another reviewers’ comments, we had changed the title of our manuscript to Estimations of Fracture Surface Area Using Tracer and Temperature Data in Geothermal Fields, which intended to convey the specific contents. We believe our manuscript satisfies your comments and hereby resubmit for your further consideration.

The detailed comments and the responses are follows.

1.

The major issue of the manuscript is the lack of a Discussion section that must be compulsory in any scientific paper. Although the methodology exposed and the results described seem to be correct, it is important to discuss the results. This section must include a description of the novelty of the method in relation to previous works, as well as to describe the weak and strong points of the proposed method. What are the benefits of this method in relation to previous ones? What are the limitations encountered? How different are the results in the two geothermal areas compared with previous works? By answering these questions, the scientific significance of the manuscript will be increased.

We added more discussion in last section to compare with other studies and other methods.

2.

Apart from this, I have some comments about the results. It is surprising that, using data form two very different geothermal systems, one artificial and another one natural, the method provides exactly the same best fit for both of them. In both cases, the best fit is obtained when the number of flow paths is nine and the initial value was 100 m2, and both of them provides the same smallest residual error of ~50º-60º C temperature. It looks like the method has a certain tendency to obtain the same fit independently of the initial model. This is something that should be commented in the Discussion section proposed.

We added discussion part and mentioned how the number of flow paths should be considered in L692.

3.

These observation results even more surprising when the authors state that, despite the fact that the best fit obtained for Fenton Hill field corresponds to a number of flow paths of nine, it is finally estimated that the best solution for this geothermal field is a single flow path, although this model has considerably greater (100º C, about twice greater) residual errors by looking at the graphs of figure 6. This is interesting because one can have serious doubts about the results of a method that assigns the lower errors to models that are not the best fit to the geothermal field characteristics, making difficult to decide how to select the best solution. This is controversial and should be clearly justified and exposed in the Discussion.

In the previous version, we used absolute errors. Instead, mean absolute error was used, which was mentioned in P12. The the number of flow paths is just a fitting parameter and not describe the real flow path. Increasing the number of flow paths increases the degree of freedom. The discussion was added in L692.

4.

At Table 1, the thickness value (6th row) ranges from 0.35 to 1.00 mm. Such an extremely small value is impossible, and it must be an error that has to be fixed.

Thank you for your notice. Fixed.

5.

At figure 3, the labels are too small and very difficult to read. Please enlarge the figure.

Enlarged it.

Round 2

Reviewer 3 Report

After carefully reading the manuscript, I consider that the revised version has beengreatly improved regarding the original submission. All my suggestions have been considered by the authors and the manuscript is now more complete and interesting for the audience. In my opinion, it can be accepted for publication.

Be careful with some minor typo errors, such as the labelling of Y axis in the graphs in figures 10, 14 and 17, or the extremely small size of the labelling (Initial value) of the X axis in figure 4, that should be fixed in print proofs stage. There is also a contradiction with the permeability value that is defined as 1.0 x 10-11 m2 in the text (lines 178-179) whereas later, in the table 1, is defined as 10-12 m2.